# Bayesian Decision Making around Experts

**Daniel Jarne Ornia**[*]
University of Oxford

**Joel Dyer**[*]
University of Oxford

**Nick Bishop**
University of Oxford

**Ani Calinescu**
University of Oxford

**Michael Wooldridge**
University of Oxford

## Abstract

Complex learning agents are increasingly deployed alongside existing experts, such as human operators or previously trained agents. However, it remains unclear how should learners optimally incorporate certain forms of expert data which may differ in structure from its own action-outcome experiences. We study this problem in the context of Bayesian multi-armed bandits, considering: (i) *offline settings*, where the learner receives a dataset of outcomes from the expert's optimal policy before interaction, and (ii) *simultaneous settings*, where the learner must choose at each step whether to update its beliefs based on its own experience, or based on the outcome simultaneously achieved by an expert. We formalize how expert data influences the learner's posterior, and prove that pretraining on expert outcomes tightens information-theoretic regret bounds by the mutual information between the expert data and the optimal action. For the simultaneous setting, we propose an information-directed rule where the learner processes the data source that maximizes their one-step information gain about the optimal action. Finally, we propose strategies for how the learner can infer when to trust the expert and when not to, safeguarding the learner for the cases where the expert is ineffective or compromised. By quantifying the value of expert data, our framework provides practical, information-theoretic algorithms for agents to intelligently decide when to learn from others.

## 1 Introduction

Many learning systems are deployed *next to other learners*: an agent learning online may co-exist with a party that already knows how to act well in the same environment (a human operator, a calibrated controller, or a previously trained policy). Examples of this include clinical decision support (learning beside clinicians), robotics (learning beside a safe supervisor) and general AI agents (small agents learning beside a powerful, well tuned large model). While Bayesian bandit algorithms and Thompson Sampling (TS) in particular offer efficient exploration strategies with information–theoretic regret guarantees [Thompson, 1933, Russo and Van Roy, 2014, 2016], it remains unclear how a Bayesian learner should optimally use expert information that differs in kind from its own action–outcome experience. Motivated by this observation, we study *online Bayesian learning next to an expert* in multi-armed bandits. In this setting, the learner interacts with a multi-armed bandit (with a prior belief over the structure of the bandit) alongside an expert who knows the true structure, and reveals the outcomes they experience. In particular, we consider two settings, (i) an *offline setting*, in which the learner has access to an offline dataset consisting of past outcomes experienced by the expert, and (ii) a *simultaneous setting*, wherein the learner acts in sync with the expert, and must choose

---

[*]{daniel.jarneornia,joel.dyer}@cs.ox.ac.uk

39th Conference on Neural Information Processing Systems (NeurIPS 2025) Workshop: Reliable ML from Unreliable Data.

between updating its beliefs in accordance with its own experience, or in accordance with the revealed outcome experienced by the expert. We investigate the following fundamental questions.

**How should the learner incorporate expert outcomes to improve decision-making?**  Intuitively, observing expert outcomes should provide tools to infer information about the optimal action distribution: The learner should be able to prune "worlds" that cannot induce that optimal action distribution. We formalize this intuition and show how to use this expert dataset to warm-start the learner's prior, and how this allows the learner to perfectly learn the environment in some instances. By leveraging these identities within the information-theoretic framework of Russo and Van Roy [Russo and Van Roy, 2016], we show that replacing the original prior with a posterior inferred from expert data yields an improved regret bound for Thompson sampling, where the degree of improvement is proportional to the reduction in entropy of the optimal action distribution.

**When learning online, which source should the learner pay attention to?**  We consider settings where at each time-step the learner can observe their own action-outcome pair, or only the expert outcome. This represents settings where agents can observe consequences of others' actions, but not necessarily which action they took. In this setting, the agent's choice of what to process is not just a matter of computational limits but of (information) opportunity cost. At each round, the decision to incorporate one piece of data means forgoing the potential knowledge gain from another. This frames the problem as one of active information source selection. The learner must then solve a deeper meta-problem: it must not only learn about the environment but also simultaneously learn who to trust; themselves or the expert. We show how this challenge (deciding whether to exploit a trusted source, explore a dubious one to test its reliability, or simply rely on self-experience) can be resolved under a unified information-theoretic framework.

**Contributions**  We introduce the problem of Bayesian online learning in the presence of experts. (i) We analyse how to incorporate expert data through a consistent Bayesian update (Proposition 1), (ii) Leveraging Russo and Van Roy [2016] we show tighter Bayesian regret bounds through an information theoretic measure of the value of expert data (Theorem 1). (iii) We propose an algorithm to choose between expert and self information by estimating the MI between $A^*$ and each source (Algorithm 1). (iv) We extend the analysis to the general case where the expert is imperfect or incorrect. (v) Finally we demonstrate how our method yields dramatic regret improvements in strongly asymmetric[2] worlds where expert outcomes nearly identify $\theta^*$.

**Main Insight**  Expert information is most valuable when it moves probability mass between optimal actions; its value can be quantified exactly by the reduction in uncertainty about the optimal action. Framing *who to learn from* as information acquisition problem about the optimal action yields both interpretable theory and practical algorithms that result in agents knowing when to (adaptively) listen to experts. This work aims to advance the theoretical understanding of settings where multiple Bayesian learners exist next to each-other with possibly different degrees of expertise or incentives, and build towards a framework for robust design of Bayesian learners in multi-agent systems.

## 1.1  Related Work

**Bandits and Beliefs**  Thompson Sampling [Thompson, 1933] has been a prevalent Bayesian algorithm for online learning for decades [Agrawal and Goyal, 2012, Chapelle and Li, 2011, Russo et al., 2018]. Russo and Van Roy [2016] made the explicit connection between the regret bounds and efficiency of Thompson Sampling and information theoretic quantities on the agent decision rules. There are also many examples of multi-agent bandit problems [Brânzei and Peres, 2021, Chang and Lu, 2025] where the question of agent information is introduced. To the best of our knowledge, these works do not consider how to incorporate expert samples in a Bayesian update and how this affects Thompson Sampling regrets. Additionally, our work traces back to early game-theoretic and theory-of-mind ideas. Works as Geanakoplos and Polemarchakis [1982], Moses and Nachum [1990] discussed the implications of agents with different belief structures sharing information to learn. Additionally, existing work on *opponent modeling* [Carmel and Markovitch, 1995, Yu et al.,

---

[2]We abuse the term symmetry to refer to problem classes where for two distinct parameters $\theta$ there exists a permutation in action labels such that the problem instances are equivalent.

2022, Nashed and Zilberstein, 2022] deals with multi-agent systems where agents model each-other's behaviour, which resonates with our ideas on learning to trust the expert.

**Learning from Experts and Demonstrations**  Our work is also connected to the broad literature on learning from expert feedback like principal-agent learning problems [Lin and Chen, 2024]. Particularly, imitation learning and inverse reinforcement learning focus on inferring a policy or reward function from an expert's actions [Abbeel and Ng, 2004, Ross et al., 2011]. Our approach differs fundamentally: we do not observe the expert's actions, but rather the outcomes generated by their known-optimal policy. This shifts the inference problem from "what did the expert do?" to "what must the world be like for the expert's policy to be optimal?". Furthermore, our setting diverges from the (frequentist) *bandits with expert advice* framework [Cesa-Bianchi et al., 1997, Auer et al., 2002], RL with expert information [Gimelfarb et al., 2018] or best action selection problems with offline data [Agrawal et al., 2023, Yang et al., 2025, Cheung and Lyu, 2024]. Here, we assume a single, observable expert, and the central point is the optimal integration of their information with the learner's Bayesian framework. Finally, our work shares motivational examples with recent work on interacting agents with different levels of expertise [Hammond and Adam-Day, 2024].

**Active Learning and Information Sources**  Our results on deciding to learn from an expert echo a form of Bayesian experimental design [Lindley, 1956] and are closely related to Information-Directed Sampling (IDS), which selects actions to optimize the trade-off between immediate reward and information gain about the optimal action [Russo and Van Roy, 2014]. However, where standard IDS chooses an action to pull, our agent makes a meta-decision about which data stream to process. This connects to Arumugam and Van Roy [2021] where the authors propose rate distortion to allow online learners to choose samples to learn from. Additionally, there are connections to recent work on regret bounds for online learning from expert feedback [Plaut et al., 2025a,b], where authors study the setting where agents can ask experts for which action is best, and work on alignment through IDS [Jeon and Van Roy, 2024]. Finally, our work connects tangentially with recent studies on poisoning datasets for Bayesian learning [Carreau et al., 2025] and how to deal with such attacks.

## 2  Single-Agent Bandit Problem

**Preliminaries**  We define our problem on a probability space $(\Omega, \mathcal{F}, P)$ with all quantities including the true parameter of the bandit, and the agent's sampled parameters, actions, and outcomes, being random variables on this space. We use lower case $x \in \mathcal{X}$ to indicate items in a set, and upper case $X$ for random variables. $\mathbb{E}[X]$ is the expected value of $X$, and the entropy of $X$ with probability mass function $p(X)$ is $\mathbb{H}(X) := -\sum_{x \in \mathcal{X}} p(x) \log p(x) = \mathbb{E}[-\log p(X)]$. The conditional entropy of $X$ given another random variable $Y$ is $\mathbb{H}(X|Y) := \mathbb{E}[-\log p(X|Y)]$, representing the remaining uncertainty in $X$ once $Y$ is known. The mutual information between $X$ and $Y$ is defined as $\mathbb{I}(X;Y) := \mathbb{H}(X) - \mathbb{H}(X|Y)$. It quantifies the reduction in uncertainty about $X$ resulting from observing $Y$. Throughout the paper, we use a subscript $t$ to denote conditioning on the history of variables up to time $t$, $\mathcal{H}_t = \{A_s, Y_s\}_{s<t}$. Furthermore, we use $P(X)$ to refer to the probability distribution of $X$, and $P(X = x)$ to refer to the probability of $X$ taking value $x$. For instance, the posterior probability of $X$ conditioned by history $\mathcal{H}_t$ is $P_t(X) := P(X \mid \mathcal{H}_t)$. Similarly, the conditional entropy of a random variable $X$ given the history is $\mathbb{H}_t(X) := \mathbb{H}(X \mid \mathcal{H}_t)$, and the conditional mutual information is $\mathbb{I}_t(X;Y) := \mathbb{I}(X;Y \mid \mathcal{H}_t)$.

**Single Agent Bandit**  An agent chooses actions $a \in \mathcal{A}$ at every time-step $t \in \mathbb{N}$, with $\mathcal{A}$ being a finite set of actions. Each action produces a (possibly random) outcome $Y_{t,a} \in \mathcal{Y}$, and the agent obtains a reward $R(Y_{t,a})$, with $R : \mathcal{Y} \to \mathbb{R}$. The outcomes are drawn from distributions $p_a$, of which the agents do not have knowledge of. We assume the outcome distribution $p_\theta := (p_{\theta,a})_{a \in \mathcal{A}}$ to be parameterised by some $\theta \in \Theta$ such that for any action, the (mean) reward is a function of $\theta$, $\mu(a, \theta) := \mathbb{E}_{Y \sim p_{\theta,a}}[R(Y)]$. Furthermore, there is a *true* parameter $\theta^*$, possibly sampled from some distribution, that defines the true bandit the agent is in. For some parameter $\theta$, the *optimal action* $a^* \in \mathcal{A}$ is then the action that satisfies $a^*(\theta) = \arg\max_{a \in \mathcal{A}} \mu(a, \theta)$. The objective of such agent is to maximize the expected cumulative reward (or equivalently, minimize the expected regret relative to the best action). The regret is defined as $Reg(T) = \sum_{t=1}^{T} R(Y_t^*) - R(Y_t)$, where $Y_t^* \sim p_{\theta, a^*}$, and we use $Y_t \equiv Y_{t,A_t}$. We will use $p^* \equiv p_{\theta^*}$, and $p_{a^*}^* \equiv p_{\theta^*, a^*(\theta^*)}$, and $\mathbb{E}[Reg(T)]$ to refer to the expected regret.

**Thompson Sampling**   Thompson sampling is a Bayesian algorithm for bandit problems that works by sampling actions according to the (posterior) probability that they are the optimal action. Let $\mathcal{H}_t := \{A_t, Y_t\}_{1 \leq t \leq T-1}$ be the history of the actions taken and outcomes observed up to (not including) time $T$. Thompson Sampling works by assuming the agent samples actions from a posterior distribution (or prior before any new observations) $P(\theta \mid \mathcal{H}_t)$ (abbreviated as $P_t(\theta)$) conditioned on $\mathcal{H}_t$ such that $P_t(A^* = a) = P_t(A_t = a)$[3]. Then, the agent samples a parameter $\hat{\theta}_t \sim P_t(\theta)$, and selects the action that maximises expected rewards *under the model* $\hat{\theta}_t$: $A_t \in \arg\max_{a \in \mathcal{A}} \mu(a, \hat{\theta}_t)$. Then, a new outcome $Y_t$ is observed (when choosing $A_t$), and the belief $P_t(\theta)$ is updated according to the history $\mathcal{H}_{t+1} = \{\mathcal{H}_t, \{A_t, Y_{A_t}\}\}$ via Bayes:

$$P_{t+1}(\hat{\theta}_t) = P(\hat{\theta}_t \mid \mathcal{H}_{t+1}) \propto p_{\hat{\theta}_t, A_t}(Y_t) P(\hat{\theta}_t \mid \mathcal{H}_t).$$

We define finally a quantity that will be of use for some of the results in the paper. From Russo and Van Roy [2016], we define the *information ratio* in a Bandit as $\Gamma_t := \mathbb{E}_t \left[ Reg(T) \right]^2 / \mathbb{I}_t(A^*, (A_t, Y_t))$. In other words, it is the ratio of the squared expected regret at time $t$ given the past history against the mutual information between the optimal action distribution and the current observation.

# 3   Learning from Expert Data

Consider the case where a *learner* has to act optimally in an unknown environment, and is able to observe an *expert* (i.e. an agent that knows $\theta^*$). This can manifest via (i) The learner gets an initial dataset $D_N^* = \{Y_n^*\}_{1 \leq n \leq N}$ and (ii) The learner gets to observe new samples $Y_t^*$ as they start learning.

## 3.1   With Expert Prior Data

**Infinite Information**   To start our analysis, assume first that $N \to \infty$ and we can construct an unbiased density estimator with no errors, or, in other words, the learner has access to the likelihood $p_{a^*}^*(Y)$. Treating this as an offline data scenario, we can interpret the knowledge of $p_{a^*}^*(Y)$ as an observation to be incorporated into the learner's knowledge via posterior inference. Intuitively, knowing $p_{a^*}^*(Y)$ should restrict the set of non-zero likelihood parameters in our posterior to those which satisfy $\tilde{\Theta} := \{\theta \in \Theta : p_{\theta, a^*(\theta)} = p_{a^*}^*\}$. Let $\mathbb{1}[p_{a^*}^* \mid \theta] = 1$ if $p_{\theta, a^*(\theta)} = p_{a^*}^*$. Then, for the posterior to be consistent with the observed data, we want it to satisfy

$$P_1(\theta \mid p_{a^*}^*) \propto P_0(\theta) \mathbb{1}[p_{a^*}^* \mid \theta]. \tag{1}$$

We use $P_0$ to refer to the initial prior the learner has over the parameters $\Theta$, and $P_1$ as the (offline) posterior resulting from incorporating the expert data. This posterior in (1) will assign zero mass[4] to any parameter $\theta$ which induces an optimal action distribution that does not match $p_{a^*}^*$. From the set of parameters that induce such a distribution, we cannot distinguish (have equal likelihood), so the prior will dominate the posterior mass. We show that this posterior update is consistent in the upcoming section, by showing it can be derived as a result of an infinite data limit.

**Finite Information**   Next, consider the case where $N < \infty$, and therefore the learner starts with a finite dataset $D_N^* = \{Y_n^*\}_{1 \leq n \leq N}$ of samples from the optimal action, but cannot identify (yet) what action these correspond to. Following the intuition in the case of infinite information, one would want to incorporate this off-line information into the prior, to afterwards proceed normally with TS, hopefully with a prior that is better informed.

Recall that, under parameter $\theta \in \Theta$, the likelihood of a given sample $Y^*$ being sampled from the bandit $\theta$ is $p_{\theta, a^*(\theta)}(Y^*)$. Then, given a set of $N$ samples $D_N^* = \{Y_n^*\}_{1 \leq n \leq N}$, we can infer a posterior under the likelihood that the data comes from the current model as

$$P_1(\theta \mid D_N^*) \propto P_0(\theta) p_{\theta, a^*(\theta)}(D_N^*). \tag{2}$$

---

[3]Note that we use $A^* = a$ to refer to the random event corresponding to $a$ being the optimal action under measure $P_t$. This differs from $a^*(\theta)$, which refers to the optimal action in expectation for a fixed $\theta$.

[4]In general, this needs to be dealt with care for compact parameter spaces and continuous measures to avoid measure theoretic issues. For this argument it suffices to assume a countable parameter set $\Theta$, and all the results extend to compact parameter sets under appropriate measure theoretic assumptions prevalent in other Bayesian bandit work.

Since the expert samples are i.i.d., we write the right hand side:

$$P_0(\theta) p_{\theta, a^*(\theta)}(D_N^*) \quad = \quad P_0(\theta) \prod_{i=1}^{N} p_{\theta, a^*(\theta)}(Y_i^*) \quad = \quad P_0(\theta) \exp\Big(\sum_{i=1}^{N} \log p_{\theta, a^*(\theta)}(Y_i^*)\Big). \quad (3)$$

**Proposition 1.** *Assume a countable set* $\Theta$. *As the number of samples increases* $N \to \infty$, *the posterior update in* (3) *converges to the infinite data update in* (1). *In other words,*

$$\lim_{N \to \infty} P_1(\theta \mid D_N^*) = P_1(\theta \mid p_{a^*}^*) \quad a.s. \tag{4}$$

**Regret Bounds with Offline Expert Data**    To estimate the Bayesian regret improvement of the agents when having access to offline expert data, let us first define the following concepts. The probability $\mathbb{P}_0(A^* = a)$ under measure $P_0$ is the probability of $a$ being optimal under the prior distribution $P_0(\theta)$. Let $\Theta_a^* := \{\theta \in \Theta : a = \arg\max_{a' \in \mathcal{A}} \mu(a', \theta)\}$; in other words, $\Theta_a^*$ is the subset of parameters that yields $a$ to be the optimal action. Observe we can then write $\mathbb{P}_0(A^* = a) = \int_{\Theta_a^*} P_0(\theta) d\theta$. Then, define $\mathbb{H}_0(A^*)$ to be the entropy of the optimal action distribution under measure $P_0$:

$$\mathbb{H}_0(A^*) = \sum_{a \in \mathcal{A}} P_0(A^* = a) \log P_0(A^* = a).$$

Russo and Van Roy [2016] established that the Bayesian regret of a Thompson Sampling algorithm is upper bounded by $\sqrt{\mathbb{H}_t(A^*)}$. We can now show that, under expert data, the entropy of the (offline) posterior $P_0(\theta \mid D_N^*)$ is guaranteed to decrease in expectation over the observed data.

**Theorem 1** (Regret Reduction from Offline Expert Data). *Let an agent follow a Thompson Sampling algorithm with a prior inferred from the expert-updated posterior* $P_1(\theta) = P(\theta \mid D_N^*)$. *Their expected Bayesian regret taken over all sources of randomness including the expert data* $D_N^*$, *is*

$$\mathbb{E}[Reg_{TS_1}(T)] \le C \sqrt{T \left(\mathbb{H}_0(A^*) - \mathbb{I}_0(A^*; D_N^*)\right)}$$

*where* $C$ *is a problem-dependent constant,* $\mathbb{H}_0(A^*)$ *is the prior entropy of the optimal action and* $\mathbb{I}_0(A^*; D_N^*)$ *is the mutual information between the optimal action and the expert data under* $P_0$.

In particular, Theorem 1 provides an upper bound on the expected regret incurred is lower than the upper bound for a TS agent who observes no expert data and assumes the same prior $P_0$ given by Russo and Van Roy [Proposition 1, 2016]. Intuitively, this means that if the mutual information between the expert data and the optimal action distribution is high (*i.e.* the expert samples allow the agent to reduce the set of possible parameters to a much smaller subset), then the resulting regret will be significantly lower.

## 3.2   Learning while observing a Trustworthy Expert

Consider now the problem where the learner has no expert data to incorporate into their prior, but as they learn, they will observe both the (action, outcome) pair $(A_t, Y_t)$ they generate themselves and the (optimal) outcome $Y_t^*$ the expert generates (and thus also knows $R(Y_t^*)$).

In this case, we assume that the learner can only learn from one sample at a time. Therefore, the learner needs to choose at every step $t$ whether they learn from the expert outcome $Y_t^*$ (which does not include actions), or their own sampled pair $(A_t, Y_t)$. We assume the learner will still receive its own reward $R(Y_t)$, and thus the expected instantaneous regret $\mathbb{E}[R(Y_t^*) - R(Y_t)]$ does not depend on the expert sample, or on the agent's choice on which information source to incorporate. This simplifies the analysis of the decision the agent needs to make. From Russo and Van Roy [2016] and Russo and Van Roy [2014], the expected regret of (general) Bayesian online learners is bounded by $\sqrt{\overline{\Gamma} \mathbb{H}_t(A^*) T}$, where $\overline{\Gamma}$ is an upper bound for the information ratio. Given that the agent's choice over what information to incorporate does not change the immediate rewards, this choice needs to be driven by the information gain from each source. Let $D_t \in \{Y_t^*, (Y_t, A_t)\}$ be the random variable representing the data processed at time $t$, which can be either the expert outcome or the pair (outcome, action) from the learner themselves. Then, the choice of data to learn from can be expressed through the choice:

$$\arg\min_{D_t} \mathbb{E}[\mathbb{H}_t(A^* \mid D_t)] \quad = \quad \arg\min_{D_t} \mathbb{H}_t(A^*) - \mathbb{I}_t(A^*, D_t) \quad = \quad \arg\max_{D_t} \mathbb{I}_t(A^*, D_t). \quad (5)$$

In other words, the agent should choose to learn from the sample that maximises the mutual information with the optimal action distribution. This is effectively a Bayesian experimental design framework [Lindley, 1956], where the *experiments* (self-generated data vs. expert data) need to be selected to maximise information gain[5].

### 3.2.1 Estimating Mutual Information

The agent's information at time $t$ is condensed in the current prior $P_t(\theta)$. In Thompson Sampling, the probability of selecting action $a$ at time $t$ (defined to be the history dependent policy $\pi_t$) is equal to the probability of $a$ being optimal,

$$P(A_t = a) = P_t(A^* = a) = \int_{\Theta_a^*} P_t(\theta)d\theta =: \pi_t(a).$$

We need to estimate the quantities $\mathbb{I}_t(A^*, Y_t^*)$ and $\mathbb{I}_t(A^*, (A_t, Y_t))$. We consider first the case where the agent cannot use the realisations $Y_t$ or $Y_t^*$ to compute information gain, but instead needs to estimate the mutual information between the random variables.

**Self-Generated Data Predictives**  The (prior) outcome marginal predictive density is

$$P_t(Y_t \mid A_t = a) = \int_{\Theta} P_t(\theta)p_{\theta,a}(Y_t)d\theta;$$

this is the marginal likelihood of observing outcome $Y_t$ having selected action $A_t = a$. Additionally, the conditional density of $Y_t$ under $A_t = a$ with the hypothesis that $A^* = a'$ is

$$P_t(Y_t \mid A_t = a, A^* = a') = \frac{P_t(Y_t, A^* = a' \mid A_t = a)}{P_t(A^* = a' \mid A_t = a)} = \frac{P_t(Y_t, A^* = a' \mid A_t = a)}{\pi_t(a')}.$$

Therefore, the MI resulting from observing self-collected data $(A_t, Y_t)$ can be computed as (see Appendix B for a derivation):

$$\mathbb{I}_t(A^*, (A_t, Y_t)) = \sum_{a \in \mathcal{A}} \pi_t(a) \sum_{a' \in \mathcal{A}} \pi_t(a')D_{KL}(P_t(Y_t \mid A_t = a, A^* = a')\|P_t(Y_t \mid A_t = a)).$$

(6)

**Expert Data Predictives**  Similarly, we can now compute predictive densities for the case where the agent only observes the expert sample $Y_t^*$, knowing it comes from the optimal action. We define the marginal predictive and the joint predictive for the expert output:

$$P_t(Y_t^*) = \int_{\Theta} P_t(\theta)p_{\theta,a^*(\theta)}(Y_t^*)d\theta, \quad P_t(Y_t^*, A^* = a') = \int_{\Theta_{a'}^*} P_t(\theta)p_{\theta,a'}(Y_t^*)d\theta.$$

We define the conditional predictive:

$$P_t(Y_t \mid A^* = a') = \frac{P_t(Y_t^*, A^* = a')}{P_t(A^* = a')} = \frac{P_t(Y_t^*, A^* = a')}{\pi_t(a')}.$$

Now, we can write the corresponding MI $\mathbb{I}_t(A^*, Y_t^*)$ as

$$\mathbb{I}_t(A^*, Y_t^*) = \sum_{a' \in \mathcal{A}} \pi_t(a')D_{KL}(P_t(Y_t^* \mid A^* = a')\|P_t(Y_t^*)).$$

(7)

**Remark 1.** *Observe that if on the contrary the agent is allowed to use the data $(A_t, Y_t)$, $Y_t^*$ to estimate information gain, one could compute the posterior action distributions $P_t(A^* \mid A_t = a, Y_t = y)$, $P_t(A^* \mid Y_t^* = y)$, and directly compute the corresponding entropies $\mathbb{H}_t(A^* \mid A_t = a, Y_t = y)$, $\mathbb{H}_t(A^* \mid Y_t^* = y)$ to select the source with the minimum entropy. However, this naturally introduces high variance from the sampling nature of the random $Y_t^*$, $Y_t$. In principle, computing the expected information gain (in the form of the MI in (7), (6)) would allow the agents to estimate the best information source with limited variance.*

---

[5]Bayesian experimental design is usually framed in terms of the information gain of model parameters $\theta$. In our case, we care about the mutual information between $(A^*, D)$.

### 3.3 When does Expert Information help?

A natural question after the results presented in previous sections is when does expert information help (and when does it not help). Recall the *expert predictives* $P_t(Y_t^* \mid A^* = a)$, $P_t(Y_t^*)$, and recall the KL mixture expression for the MI in (7). We can quickly establish the following result.

**Proposition 2.** *The mutual information* $\mathbb{I}_t(A^*; Y_t^*) = 0$ *if and only if* $P_t(Y_t^* \mid A^* = a) = P_t(Y_t^*)$ *for every* $a \in \mathcal{A}$ *i.e., if and only if* $P_t(Y_t^* \mid A^* = a)$ *is identical across optimal-action labels* $a$.

**Corollary 1** (Symmetric Worlds)**.** *If the optimal-action likelihood is the same in every world,* $p_{\theta, a^*(\theta)}(\cdot) \equiv q(\cdot)$ *for all* $\theta \in \Theta$ *and some* $q \in \Delta(\mathcal{Y})$, *then* $\mathbb{I}_t(A^*; Y_t^*) = 0$ *for any* $P_t$. *In other words, the expert data leaves the posterior unchanged.*

**When can** $\mathbb{I}_t(A^*; Y_t^*) > 0$**?**     From Proposition 2, expert outcomes help exactly when the action-indexed measures $P_t(Y_t^* \mid A^* = a)$ differ. This happens when the *posterior breaks symmetry* across $\{\Theta_a^*\}_a$ (e.g. due to asymmetric priors or asymmetric self-collected data), or the model family is only *weakly symmetric* so that $p_{\theta, a}$ varies within each $\Theta_a^*$ and the induced probabilities $P_t(Y_t^* \mid A^* = a)$ are different under exchangeable beliefs. Then, $Y_t^*$ moves probability mass between hypotheses $\{A^* = a\}_a$, lowering $\mathbb{H}_t(A^*)$ and improving regret bounds via Theorem 1. For a generalisation to problems where the learner needs to model the expert behaviour, see Appendix C.3.

## 4 Experiments

We present now a set of bandit experiments to showcase the results presented in previous sections. We fix all experiments to $\mathcal{Y} = \{-50, ..., 50\}$ and $R(Y) = Y$ is the identity map and unless specifically stated, $\text{supp}(p_{\theta, a}) = \mathcal{Y}$ for all $\theta, a$.

**Symmetric Worlds:**     Countable[6] $\Theta = \{\theta_m\}_{m \leq M}$ where all bandits have the same set of actions $\mathcal{A}$ with finite supports, but *shuffled*. That is, each bandit will have the same optimal action distribution assigned to a different action. In this case, there is no information gain from expert data.

**Asymmetric Worlds:**     Countable $\Theta = \{\theta_m\}_{m \leq M}$ where all bandits have the same number of actions with equal support, but the probability distributions $p_{\theta, a}$ are generated at random for each $\theta, a$ by adding normally distributed noise to a uniform distribution. That is, every bandit has (similar but) numerically different action distributions. In this case, using expert data should asymptotically lead to zero regret.

**Strongly Asymmetric Worlds:**     Countable $\Theta = \{\theta_m\}_{m \leq M}$ where all bandits have the same number of actions with equal support, the probability distributions $p_{\theta, a}$ are generated at random for each $\theta, a$, but we fix the true bandit $\theta^*$ to have $p_{A^*}^*(y^*) = 1$ for some fixed $y^*$ with positive reward. On average, this problem is similarly hard to a traditional Thompson Sampling agent, but an agent learning from expert data should infer with few samples the true $\theta^*$.

### 4.1 Symmetric Bandits

We present first the learning results on the symmetric bandits with countable parameter set. We generate $M = 500$ bandit models (parameters) by generating 50 actions from adding random noise to a uniform distribution over $\mathcal{Y}$ and normalizing. Then, we select one model at random from the 500 parameters to be the true model. The prior is $P_0(\theta) = \text{uniform}(\Theta)$ in all cases. We run each scenario with 50 different random seeds and present all runs in transparent color, and the means in thicker opaque lines. In all cases, we plot the cumulated regret rate $Reg(T)/T$.

**Results in Symmetric Bandits**     The results are presented in Figure 1. First, we can see how offline learning with expert samples does not improve the Thompson Sampling regret at all in the symmetric bandit case. Having information over the optimal action distribution does not help when all bandits for any $\theta$ have the same optimal action distribution. Second, the fastest learning rate is obtained for the case where the agent only considers their own data at every time-step. Learning from expert data

---

[6]We restrict the experiments to countable worlds and finite actions since this allows us to express priors and posteriors with categorical distributions and compute Bayesian updates exactly.

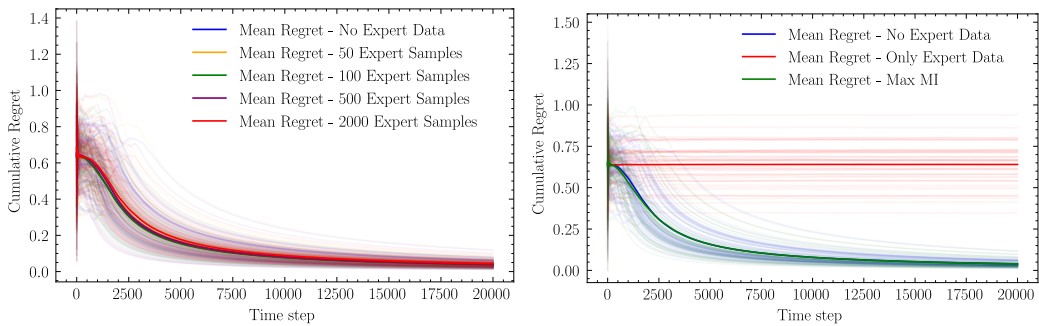

Figure 1: Regret obtained by TS agents with expert data in symmetric bandits. Left: Pretraining with expert samples. Right: Selecting information sources.

only results in linear regret (no learning). Interestingly, the MI estimating agent is able to discriminate the sources and consistently chooses to learn from its own data, successfully filtering out useless information.

## 4.2 Asymmetric Bandits

We simulate agents with $M = 500$ bandits and $|\mathcal{A}| = 50$ where for each $\theta$ the distributions $p_{a,\theta}$ are generated as a (renormalised) uniform distribution over $\mathcal{Y}$ with Gaussian zero mean noise in each entry. This results in bandit instances that are hard to distinguish, but that have different distributions for each $a, \theta$.

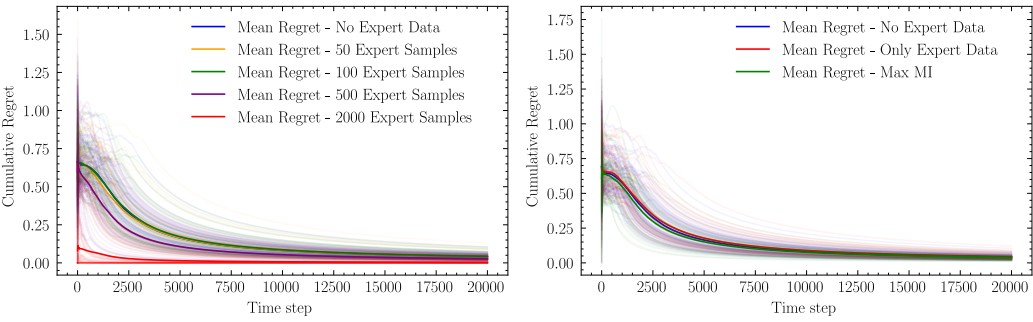

Figure 2: Regret obtained by TS agents with expert data in asymmetric bandits. Left: Pretraining with expert samples. Right: Selecting information sources.

**Results in Asymmetric Bandits**    We present the corresponding results in Figure 2. In this case, we can observe how having access to an expert dataset offline yields heavy improvements in total regret when running Thompson Sampling with the resulting posteriors. In the case with 2000 expert samples, the resulting agents achieve almost zero regret from the start of the Thompson Sampling phase. Interestingly, in this case the selection of information source does result in an overall improvement in learning speed. In particular, when comparing the regret rate at low time-steps ($t = 2000$), the agents running Algorithm 1 get an improvement of $-12\%$ and $-8\%$ correspondingly with respect to single source agents. These values may seem moderate, but they are in fact quite significant considering the overall setting. It means that, across a wide range of randomly generated problem instances, selecting information sources based on past data results in a $\approx 10\%$ learning rate improvement over an (already efficient) Thompson Sampling agent at no additional sampling cost.

## 4.3 Strongly Asymmetric Bandits

To test the cases where having expert data solves the bandit problem almost immediately, we simulated agents with $M = 500$ and $|\mathcal{A}| = 50$ bandits, with distributions generated identically to the previous asymmetric experiments, but with one change. Once the true parameter $\theta^*$ is selected (at random),

one of the action distributions $a'$ is replaced by a (Dirac delta) distribution $p_{a', \theta^*}(2) = 1$. This results in all cases in $a' = A^*$. Since the agent knows the problem class, solving the bandit in a traditional Thompson Sampling approach will still require a large amount of steps, but having expert samples would allow the agent to immediately infer $\theta^*$.

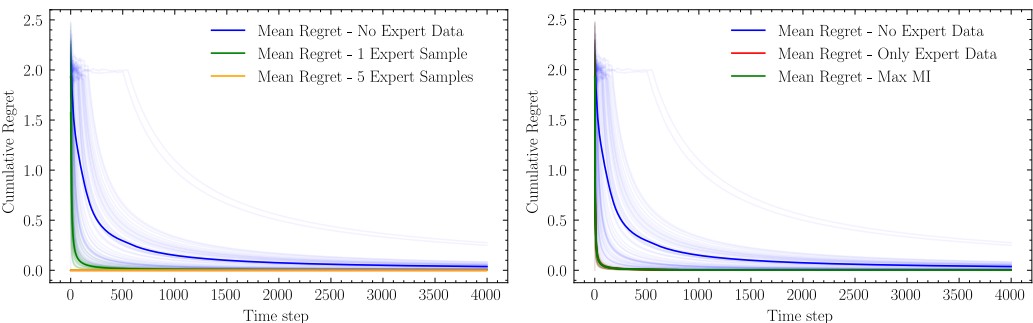

Figure 3: Regret obtained by TS agents with expert data in strongly asymmetric bandits. Left: Pretraining with expert samples. Right: Selecting information sources.

**Results in Strongly Asymmetric Bandits**    The results are presented in Figure 3. Observe that, in the left hand plot, having a single expert sample to compute an offline prior causes the regret rate to drop almost immediately after a few Thompson Sampling steps. For only 5 expert samples, the resulting offline posterior yields a zero regret Thompson Sampling algorithm for all times in all instances computed. In this case, the improvements in regret rates are dramatic for agents running Algorithm 1. In particular, for agents using a single sample to estimate the MI, after 500 steps the improvement in regret is of $-99\%$ when compared to regular Thompson Sampling. This means the agents are successfully able to estimate that the gains in mutual information from the expert source are very beneficial and choose to learn from this source.

# 5    Discussion

In this work we studied the problem of Bayesian online learning when agents have access to expert *outcomes*, and are able to use these outcomes to improve their learning. We circle back now to the fundamental questions in the introduction.

**On how to incorporate expert outcomes**    We showed how expert information is most valuable in worlds where it provides discriminative evidence to prune the parameter space. In such settings, both offline pre-training and our simultaneous learning algorithm dramatically reduce regret, and we showed our information theoretic framework results in consistent, optimal learning from expert data. Agents reduce their regret by exactly the amount of useful information present in the expert data.

**On what source to pay attention to**    We showed theoretically and empirically how estimating the MI from each information source has a direct impact in the expected regret obtained and is an appropriate metric for this meta-decision making process. Our extension to untrustworthy experts addresses a critical robustness gap in agents that learn from external sources. Expert outcomes serve a dual informational role: they provide evidence about the world's underlying parameters while simultaneously serving as a check on the agent's belief in the expert's reliability. Our proposed MI-based decision rule provides a principled mechanism for navigating this trade-off. This transforms the problem from simple learning-beside-an-expert to a more realistic and general challenge of learning how to learn in a world with multiple, imperfect information sources.

**Limitations and Future Work**    Parts of our analysis were conducted in countable parameter and action spaces, which enabled exact posterior updates. Extending this framework to continuous spaces with function approximation is a significant next step, as well as extending the general problem class to state-based Markov Decision Processes. Most critically, future work will include simultaneous learning settings, where the expert's policy is not static, but is evolving as the expert learns from their own experiences too.

## Acknowledgements

Authors would like to thank David Hyland, Wei-Chen Lee, Lewis Hammond and Christian Schroeder de Witt for the useful conversations on the topic. Authors acknowledge funding from a UKRI AI World Leading Researcher Fellowship awarded to Wooldridge (grant EP/W002949/1). MW and AC also acknowledge funding from Trustworthy AI - Integrating Learning, Optimisation and Reasoning (TAILOR), a Horizon2020 research and innovation project (Grant Agreement 952215).

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

# A  Mathematical Proofs

*Proposition 1.* First, let us write

$$\sum_{i=1}^{N} \log p_{\theta,a^*(\theta)}(Y_i^*) = N\Big(\frac{1}{N}\sum_{i=1}^{N} \log p_{\theta,a^*(\theta)}(Y_i^*)\Big).$$

Now we have

$$\frac{1}{N}\sum_{i=1}^{N} \log p_{\theta,a^*(\theta)}(Y_i^*) = \mathbb{E}_{Y\sim p_{a^*}^*}[\log p_{\theta,a^*(\theta)}(Y)] + \delta_N(\theta),$$

and $\delta_N(\theta) := \frac{1}{N}\sum_{i=1}^{N} \log p_{\theta,a^*(\theta)}(Y_i^*) - \mathbb{E}_{Y\sim p_{a^*}^*}[\log p_{\theta,a^*(\theta)}(Y)]$, which goes to zero *almost surely* as $N \to \infty$ by the law of large numbers. Observe that $\mathbb{E}_{Y\sim p_{a^*}^*}[\log p_{\theta,a^*(\theta)}(Y)]$ is the cross entropy between $p_{a^*}^*$ and $p_{\theta,a^*(\theta)}$, and thus

$$\mathbb{E}_{Y\sim p_{a^*}^*}[\log p_{\theta,a^*(\theta)}(Y)] = -\mathbb{H}(p_{a^*}^*) - D_{KL}(p_{a^*}^*||p_{\theta,a^*(\theta)}).$$

Then, substituting back in the posterior update,

$$P_1(\theta \mid D_N^*) = \frac{\prod_{i=1}^{N} p_{\theta,a^*(\theta)}(Y_i^*)P_0(\theta)}{\sum_{\nu\in\Theta}\prod_{i=1}^{N} p_{\nu,a^*(\nu)}(Y_i^*)P_0(\nu)} =$$

$$\frac{\exp\big(\sum_{i=1}^{N}\log p_{\theta,a^*(\theta)}(Y_i^*)\big)P_0(\theta)}{\sum_{\nu\in\Theta}\exp\big(\sum_{i=1}^{N}\log p_{\nu,a^*(\nu)}(Y_i^*)\big)P_0(\nu)} =$$

$$= \frac{\exp\big(N\big(\mathbb{E}_{Y\sim p_{a^*}^*}[\log p_{\theta,a^*(\theta)}(Y)] + \delta_N(\theta)\big)\big)P_0(\theta)}{\sum_{\nu\in\Theta}\exp\big(N\big(\mathbb{E}_{Y\sim p_{a^*}^*}[\log p_{\nu,a^*(\nu)}(Y)] + \delta_N(\nu)\big)\big)P_0(\nu)} =$$

$$= \frac{\exp\big(N\big(-\mathbb{H}(p_{a^*}^*) - D_{KL}(p_{a^*}^*||p_{\theta,a^*(\theta)}) + \delta_N(\theta)\big)\big)P_0(\theta)}{\sum_{\nu\in\Theta}\exp\big(N\big(-\mathbb{H}(p_{a^*}^*) - D_{KL}(p_{a^*}^*||p_{\nu,a^*(\nu)}) + \delta_N(\nu)\big)\big)P_0(\nu)} =$$

$$= \frac{\exp\big(N\big(-D_{KL}(p_{a^*}^*||p_{\theta,a^*(\theta)}) + \delta_N(\theta)\big)\big)P_0(\theta)}{\sum_{\nu\in\Theta}\exp\big(N\big(-D_{KL}(p_{a^*}^*||p_{\nu,a^*(\nu)}) + \delta_N(\nu)\big)\big)P_0(\nu)},$$

where the last step holds since $\exp(-\mathbb{H}(p_{a^*}^*))^N$ does not depend on $\theta$ and it cancels out with the normalization constant.

From the definition of almost sure convergence we have that, for any $\epsilon > 0$ and for almost every $\omega \in \Omega$, there exists a $0 < N_{\omega,\epsilon}' < \infty$ such that for all $N > N_{\omega,\epsilon}'$ we have $|\delta_N(\theta)| \leq \epsilon$. Notice that in particular this means that for any $\epsilon_0 \in (0,1)$ and almost every $\omega \in \Omega$ there exists an $0 < N_{\omega,\epsilon}' < \infty$ with $\epsilon = \epsilon_0 \cdot \min_{\theta'\in\Theta\setminus\tilde{\Theta}} D_{KL}(p_{a^*}^*||p_{a^*,\theta'}) > 0$ such that $|\delta_N(\theta)| \leq \epsilon$ for all $N > N_{\omega,\epsilon}'$. Finally, this implies that for almost any $\omega \in \Omega$ and for this $\epsilon$, we have for $N > N_{\omega,\epsilon}'$

$$\begin{aligned}0 &\leq \exp\big(-D_{KL}(p_{a^*}^*||p_{a^*,\theta'}) + \delta_N(\theta)\big)\\ &\leq \exp\big(-D_{KL}(p_{a^*}^*||p_{a^*,\theta'}) + |\delta_N(\theta)|\big)\\ &\leq \exp\big(-[1-\epsilon_0]D_{KL}(p_{a^*}^*||p_{\theta,a^*(\theta)})\big)\\ &< 1.\end{aligned} \tag{8}$$

Thus since $\lim_{N\to\infty}\big(\exp\big(-[1-\epsilon_0]D_{KL}(p_{a^*}^*||p_{\theta,a^*(\theta)})\big)\big)^N \to 0$ we also have almost surely that

$$\lim_{N\to\infty}\big(\exp\big(-D_{KL}(p_{a^*}^*||p_{a^*,\theta'}) + \delta_N(\theta)\big)\big)^N \to 0.$$

Now, let us consider the subsets $\tilde{\Theta}$ and $\Theta \setminus \tilde{\Theta}$. First, take $\theta \in \tilde{\Theta}$. For any such theta, the posterior update $\forall\,\theta \in \tilde{\Theta}$ is

$$\lim_{N\to\infty} P_1(\theta \mid D_N^*) =$$

$$= \lim_{N\to\infty}\frac{\exp\big(N\delta_N(\theta)\big)P_0(\theta)}{\sum_{\nu\in\Theta}\exp\big(N\big(-D_{KL}(p_{a^*}^*||p_{\nu,a^*(\nu)}) + \delta_N(\nu)\big)\big)P_0(\nu)}.$$

Dividing the numerator and denominator by $\exp\left(N\delta_N(\theta)\right)$,

$$\lim_{N\to\infty} P_1(\theta \mid D_N^*) =$$

$$= \lim_{N\to\infty} \frac{P_0(\theta)}{\sum_{\nu\in\Theta} \exp\left(N\left(-D_{KL}(p_{a^*}^*\|p_{\nu,a^*(\nu)}) + \delta_N(\nu) - \delta_N(\theta)\right)\right)P_0(\nu)}.$$

First, any term in the denominator with $\nu \in \tilde{\Theta}$ has the same likelihood function for the optimal action. Therefore, $\delta_N(\nu) - \delta_N(\theta) = 0$ $a.s.$ for any $\nu, \theta \in \tilde{\Theta}$. Second, by the same argument as (8), any term $\nu \notin \tilde{\Theta}$ goes to zero. Therefore,

$$\lim_{N\to\infty} \frac{P_0(\theta)}{\sum_{\nu\in\Theta} \exp\left(N\left(-D_{KL}(p_{a^*}^*\|p_{\nu,a^*(\nu)}) + \delta_N(\nu) - \delta_N(\theta)\right)\right)P_0(\nu)} =$$

$$= \frac{1}{\sum_{\nu\in\tilde{\Theta}} P_0(\nu)} P_0(\theta) \quad \forall\, \theta \in \tilde{\Theta}.$$

Now consider $\theta \in \Theta \setminus \tilde{\Theta}$. Pick an arbitrary reference $\nu_0 \in \tilde{\Theta}$. We can bound the limit fraction as:

$$\lim_{N\to\infty} \frac{\exp\left(N\left(-D_{KL}(p_{a^*}^*\|p_{\theta,a^*(\theta)}) + \delta_N(\theta)\right)\right)P_0(\theta)}{\sum_{\nu\in\Theta} \exp\left(N\left(-D_{KL}(p_{a^*}^*\|p_{\nu,a^*(\nu)}) + \delta_N(\nu)\right)\right)P_0(\nu)} \le$$

$$\le \lim_{N\to\infty} \frac{\exp\left(N\left(-D_{KL}(p_{a^*}^*\|p_{\theta,a^*(\theta)}) + \delta_N(\theta)\right)\right)P_0(\theta)}{\exp\left(N\delta_N(\nu_0)\right)P_0(\nu_0)} \quad \forall\, \theta \in \Theta \setminus \tilde{\Theta}.$$

Now, re-arranging terms,

$$\lim_{N\to\infty} \frac{\exp\left(N\left(-D_{KL}(p_{a^*}^*\|p_{\theta,a^*(\theta)}) + \delta_N(\theta)\right)\right)P_0(\theta)}{\exp\left(N\delta_N(\nu_0)\right)P_0(\nu_0)} =$$

$$= \lim_{N\to\infty} \exp\left(N\left(-D_{KL}(p_{a^*}^*\|p_{\theta,a^*(\theta)}) - \delta_N(\nu_0) + \delta_N(\theta)\right)\right)\frac{P_0(\theta)}{P_0(\nu_0)}.$$

By the same argument as (8), the exponent limit goes to zero, and thus $\forall\, \theta \in \Theta \setminus \tilde{\Theta}$:

$$\lim_{N\to\infty} \frac{\exp\left(N\left(-D_{KL}(p_{a^*}^*\|p_{\theta,a^*(\theta)}) + \delta_N(\theta)\right)\right)P_0(\theta)}{\sum_{\nu\in\Theta} \exp\left(N\left(-D_{KL}(p_{a^*}^*\|p_{\nu,a^*(\nu)}) + \delta_N(\nu)\right)\right)P_0(\nu)} \le 0.$$

This completes the proof, and we have

$$\lim_{N\to\infty} P_1(\theta \mid D_N^*) = \frac{\mathbb{I}[p_{a^*}^* \mid \theta]}{\sum_{\nu\in\tilde{\Theta}} P_0(\nu)} P_0(\theta).$$

$\square$

*Theorem 1 (Regret Reduction from Offline Expert Data).* The result follows directly from the information-theoretic analysis of Russo and Van Roy [2016], which bounds the Bayesian regret of a Thompson Sampling agent by the entropy of the optimal action under its current belief distribution. The agent has belief $P_1$. The regret, conditioned on a specific realization of $D_N^*$, is bounded by:

$$\mathbb{E}[Reg(T) \mid D_N^*] \le C\sqrt{T \cdot \mathbb{H}_1(A^*)}$$

To find the unconditional expected regret, we take the expectation over the expert data $D_N^* \sim p_{a^*}^*(\theta^*)$, where the uncertainty about $\theta^*$ is captured by the prior $P_0$:

$$\mathbb{E}[Reg(T)] = \mathbb{E}_{D_N^*}\left[\mathbb{E}[Reg(T) \mid D_N^*]\right] \le$$

$$\le \mathbb{E}_{D_N^*}\left[C\sqrt{T \cdot \mathbb{H}_1(A^*)}\right].$$

Applying Jensen's inequality we have $\mathbb{E}[\sqrt{X}] \le \sqrt{\mathbb{E}[X]}$, which gives:

$$\mathbb{E}[Reg(T)] \le C\sqrt{T \cdot \mathbb{E}_{D_N^*}[\mathbb{H}_1(A^*)]}.$$

The entropy $\mathbb{H}_1(A^*)$ is precisely the conditional entropy of the optimal action given the expert data, under the original measure $P_0$:

$$\mathbb{H}_1(A^*) = -\sum_{a \in \mathcal{A}} P_1(A^* = a) \log P_1(A^* = a) =$$

$$= -\sum_{a \in \mathcal{A}} P_0(A^* = a \mid D_N^*) \log P_0(A^* = a \mid D_N^*) =: \mathbb{H}_0(A^* \mid D_N^*).$$

Substituting this into the bound, we get:

$$\mathbb{E}[Reg(T)] \leq C \sqrt{T \cdot \mathbb{E}_{D_N^*}[\mathbb{H}_0(A^* \mid D_N^*)]}.$$

Finally, from the definition of mutual information: $\mathbb{E}_{D_N^*}[\mathbb{H}_0(A^* \mid D_N^*)] = \mathbb{H}_0(A^*) - \mathbb{I}_0(A^*; D_N^*)$. This yields the main result:

$$\mathbb{E}[Reg(T)] \leq C \sqrt{T \left(\mathbb{H}_0(A^*) - \mathbb{I}_0(A^*; D_N^*)\right)}.$$

This completes the proof. $\qquad\square$

*Proposition 2.* By (7), $\mathbb{I}_t(A^*; Y_t^*) = 0$ if and only if each $D_{KL}(P_t(Y_t^* \mid A^* = a) \mid P_t(Y_t^*)) = 0$, which holds if and only if $P_t(Y_t^* \mid A^* = a) = P_t(Y_t^*)$ for all $a$. $\qquad\square$

*Corollary 1.* Assume that for every $\theta \in \Theta$,

$$p_{\theta, a^*(\theta)}(y) \equiv q(y) \qquad \text{for all } y \in \mathcal{Y}, \tag{9}$$

for some fixed density function $q$ on $\mathcal{Y}$. We now prove (i) $A^*$ and $Y_t^*$ are independent under $P_t$ so $\mathbb{I}_t(A^*; Y_t^*) = 0$, and (ii) an expert sample leaves the posterior over $\theta$ unchanged.

First, for any action $a$ and any measurable $B \subseteq \mathcal{Y}$,

$$P_t(Y_t^* \in B \mid A^* = a) =$$

$$= \int_{\Theta_a^*} \left(\int_B p_{\theta, a}(y)\, dy\right) P_t(\theta \mid A^* = a)\, d\theta = \int_B q(y)\, dy =: Q(B),$$

which does not depend on $a$. Note that the first equality comes from the definition of the conditional measure $P_t(Y_t^* \in B \mid A^* = a)$, and the second equality comes from assumption (9) and from the fact that $\Theta_a^*$ is the subset of $\Theta$ that includes all $\theta$ with $A^* = a$. Hence $P_t(Y_t^* \mid A^* = a) = Q$ for all $a$, and the (unconditional) predictive is also $P_t(Y_t^*) = \sum_a \pi_t(a) Q(Y_t^*) = Q(Y_t^*)$. From (7),

$$\mathbb{I}_t(A^*; Y_t^*) = \sum_{a \in \mathcal{A}} \pi_t(a)\, D_{\mathrm{KL}}\big(P_t(Y_t^* \mid A^* = a) \,\|\, P_t(Y_t^*)\big), \tag{10}$$

and each term equals $D_{\mathrm{KL}}(Q \| Q) = 0$, hence $\mathbb{I}_t(A^*; Y_t^*) = 0$.

Now let $y^*$ be an observed expert outcome. Bayes' rule gives, for countable $\Theta$ (the general case follows by replacing sums with integrals),

$$P_{t+1}(\theta \mid Y_t^* = y^*) \propto p_{\theta, a^*(\theta)}(y^*)\, P_t(\theta) = q(y^*)\, P_t(\theta). \tag{11}$$

After normalizing by $\sum_{\vartheta} q(y^*) P_t(\vartheta) = q(y^*)$, we obtain $P_{t+1}(\theta \mid y^*) = P_t(\theta)$. This completes the proof. $\qquad\square$

## B How to select Information Sources

### B.1 Deriving the Predictive Densities

For any $a' \in \mathcal{A}$, the joint density of $(A^*, Y_t)$ conditional on $A_t = a$ is

$$P(Y_t, A^* \mid A_t = a) = \int_{\Theta_{A^*}^*} P_t(\theta) p_{\theta, a}(Y_t) d\theta,$$

and observe that

$$P_t(Y_t \mid A_t = a) = \sum_{a' \in \mathcal{A}} P(Y_t, A^* = a' \mid A_t = a).$$

This density indicates how much of the probability mass of any given outcome $y$ comes from the worlds where $A^* = a'$. Finally, the conditional density of $Y_t$ under $A_t = a$ with the hypothesis that $A^* = a'$ is

$$P_t(Y_t \mid A_t = a, A^* = a') = \frac{P_t(Y_t, A^* = a' \mid A_t = a)}{P_t(A^* = a' \mid A_t = a)} = \frac{P_t(Y_t, A^* = a' \mid A_t = a)}{\pi_t(a')}.$$

Now observe, from the definition of MI and since the history dependent policy does not depend on $\theta$, the choice of $A_t$ carries no information about $A^*$ so we can write:

$$\mathbb{I}_t(A^*, (A_t, Y_t)) = \mathbb{I}_t(A^*, Y_t \mid A_t) = \sum_{a \in \mathcal{A}} \pi_t(a) \mathbb{I}_t(A^*, Y_t \mid A_t = a).$$

Furthermore we marginalise over actions and write:

$$\mathbb{I}_t(A^*, Y_t \mid A_t = a) = \sum_{a' \in \mathcal{A}} P_t(A^* = a') D_{KL}(P_t(Y_t \mid A^* = a') \| P_t(Y)) =$$

$$= \sum_{a' \in \mathcal{A}} \pi_t(a') D_{KL}(P_t(Y_t \mid A_t = a, A^* = a') \| P_t(Y_t \mid A_t = a)).$$

Therefore, the MI resulting from observing self-collected data $(A_t, Y_t)$ can be computed as:

$$\mathbb{I}_t(A^*, (A_t, Y_t)) = \sum_{a \in \mathcal{A}} \pi_t(a) \sum_{a' \in \mathcal{A}} \pi_t(a') D_{KL}(P_t(Y_t \mid A_t = a, A^* = a') \| P_t(Y_t \mid A_t = a)). \tag{12}$$

---

**Algorithm 1** Information Choice: Who To Learn From

---

1: Sample particles $\{(\theta^{(n)}, w_n)\}_{n=1}^K$
2: Compute $\pi_t(a)$
3: Initialize $I_e \leftarrow 0$, $I_s \leftarrow 0$
4: **for** $l = 1, \dots, L$ **do**
5:     Sample $y_e \sim P_t(Y_t^*)$ and $y_a \sim P_t(Y_t \mid A_t = a)$ for $a \in \mathcal{A}$.
6:     Compute $P_t(Y_t^* = y_e)$, $P_t(Y_t^* = y_e, A^* = a')$, $P_t(Y_t = y_a \mid A_t = a)$, $P_t(Y_t = y_a, A^* = a' \mid A_t = a)$
7:     Set $P(A^* = a' \mid y_e) = \frac{P_t(Y_t^* = y_e, A^* = a')}{P_t(Y_t^* = y_e)}$, $P(A^* = a' \mid a, y_a) = \frac{P_t(Y_t = y_a, A^* = a' \mid A_t = a)}{P_t(Y_t = y_a \mid A_t = a)}$.
8:     $I_e^{(l)} = \sum_{a'} P(A^* = a' \mid y_e) \ln \frac{P_t(Y_t^* = y_e \mid A^* = a')}{P_t(Y_t^* = y_e)}$
9:     $I_s^{(l)} = \sum_a \pi_t(a) \sum_{a'} P(A^* = a' \mid a, y_a) \ln \frac{P_t(Y_t = y_a \mid A^* = a', A_t = a)}{P_t(Y_t = y_a \mid A_t = a)}$.
10:     $I_e \leftarrow I_e + I_e^l$, $I_s \leftarrow I_s + I_s^l$
11: **end for**
12: $I_e \leftarrow I_e/L$,    $I_s \leftarrow I_s/L$
13: **Decision:** $d_t \in \arg\max_{d \in \{e,s\}} \{I_e, \; I_s\}$

---

## C   Imperfect Experts

Until now, the analysis has focused on the setting in which the learning agent *fully trusts the expert*; there is an implicit assumption that expert samples are drawn (with full confidence) from the optimal action distribution $p_{a^*}^*$. A natural question that follows is how this can be affected by misaligned, imperfect, or adversarial experts. This can introduce robustness failure modes in agent learning, some of which can be more severe than others. To address this, we first discuss the impact of imperfect experts in the current framework, and propose afterwards a strategy to incorporate expert trust in an online Bayesian learning agent.

## C.1 Collapse under Naive Expert Trust

Consider the case where expert is sampling and providing outcomes using some (possibly adversarial) policy $\pi_e^* \in \Delta(\mathcal{A})$[7]. First, let us define $q \in \Delta(\mathcal{Y})$ as the marginal likelihood of outcomes induced by $\pi^e$: $q(Y) := \sum_{a \in \mathcal{A}} \pi_e(a) p_a^*(Y)$. Take $N$ samples from $q$, $\{Y_n^e \sim q\}_{1 \leq N}$. Recall that, since the learner is naive, it still updates its posterior based on the data:

$$P_1^q(\theta) \propto P_0(\theta) \prod_{n=1}^{N} p_{\theta, a^*(\theta)}(Y_n^q) = P_0(\theta) \exp\left(\sum_{n=1}^{N} \log p_{\theta, a^*(\theta)}(Y_n^q)\right). \qquad (13)$$

Observe that this is a specific form of a misspecified Bayesian inference problem; the agent is trying to infer a posterior thinking the data is coming from $p_{\theta, a^*(\theta)}$, and uses a corresponding likelihood, while the data is in fact sampled from a different distribution [Nott et al., 2023]. Let us use $l_N^q(\theta) := \frac{1}{N} \sum_{n=1}^{N} \log p_{\theta, a^*(\theta)}(Y_n^q)$, and $\delta_N(\theta) := \frac{1}{N} \sum_{i=1}^{N} \log p_{\theta, a^*(\theta)}(Y_i^*) - \mathbb{E}_{Y \sim p_{a^*}^*}[\log p_{\theta, a^*(\theta)}(Y)]$, and observe that $l_N^q(\theta) = \mathbb{H}(q) - D_{KL}(q \| p_{\theta, a^*(\theta)}) + \delta_N^q(\theta)$. The optimal action distribution under the misspecified posterior $P_1^q$ is[8]

$$P_1^q(a = A^*) = \frac{\int_{\theta \in \Theta_a^*} P_0(\theta) e^{N l_N^q(\theta)} d\theta}{\sum_{b \in \mathcal{A}} \int_{\theta \in \Theta_b^*} P_0(\theta) e^{N l_N^q(\theta)} d\theta} = \frac{\int_{\theta \in \Theta_a^*} P_0(\theta) e^{N(-D_{KL}(q \| p_{\theta, a^*(\theta)}) + \delta_N^q(\theta))} d\theta}{\sum_{b \in \mathcal{A}} \int_{\theta \in \Theta_b^*} P_0(\theta) e^{N(-D_{KL}(q \| p_{\theta, a^*(\theta)}) + \delta_N^q(\theta))} d\theta}.$$

For $N \to \infty$, from established misspecified Bayes results [Berk, 1966, Bochkina, 2019] and under mild assumptions (measurability, compact $\Theta_a^*$, $P_0(\theta) > 0$...) the posterior $P_1^q(a = A^*)$ will concentrate probability mass around the set $\Theta_q := \{\theta \in \Theta : \min_\theta D_{KL}(q \| p_{a^*(\theta), \theta})\}$; in other words, the set of parameters that result in an optimal action distribution that is as close as possible to $q$. We discuss two possible scenarios.

**The expert agent is Boundedly Rational** The simplest example of robustness failure is the case where the expert agent provides samples using a *boundedly rational policy*; The expert policy $\pi_e(a^*(\theta^*)) = 1 - \epsilon$ assigns some mass to the true optimal action under $\theta^*$, and some mass $\epsilon$ to the other actions. The asymptotic effect on the offline posterior $P_1^q$ will depend on the specific problem instance. If $\epsilon$ is small, then $\theta^*$ still be the minimiser $\theta^* = \min_\theta D_{KL}(q \| p_{a^*(\theta), \theta})$. In this case, the posterior will still concentrate around $\theta^*$ asymptotically and the agent will learn in the limit, but at a slower rate. For an empirical example on this, see Appendix C.

**The expert agent is adversarial** A more aggressive example is one where the expert is adversarial (and possibly deceptive), and samples with probability $\epsilon \in [0, 1]$ a true optimal outcome from $p_A^*$, and with probability $1 - \epsilon$ an *adversarial outcome* that steers the agent's beliefs over $\theta$ to the *worse possible parameter* (this is know as the Huber contamination model [Huber, 1992]). In other words, the parameter $\theta^{adv} \in \Theta$ such that $\theta^{adv} := \min_{\theta \in \Theta} \mu(a^*(\theta), \theta^*)$. In this case, depending on the problem instance, there is a threshold $\epsilon^*$ after which the agent will inevitably incur linear regret; whenever $D_{KL}((1-\epsilon)p_{a^*}^* + \epsilon p_{a^*}^* \| p_{a^*}^*) \leq D_{KL}((1-\epsilon)p_{a^*}^* + \epsilon p_{a^*}^* \| p_{a^*(\theta^{adv})}^*)$, the agent will end up being *confidently wrong*. For an empirical example on this, see Appendix C.

## C.2 Experiments on Adversarial Experts

We include here empirical results on the adversarial cases described in Section C.1. We use the same asymmetric countable world setting as in Section 4. We compute results for the following:

- A scenario with a 'mistaken' expert, where the expert samples with probability $\epsilon$ a true optimal outcome and samples with probability $1 - \epsilon$ an outcome from a uniform action distribution over $\mathcal{A} \setminus A^*$.

- A scenario with an 'adversarial' expert, where the expert samples with probability $\epsilon$ a true optimal outcome and samples with probability $1 - \epsilon$ an outcome from an optimal action in an adversarial world $\theta^{adv}$.

---

[7]This is a generalisation over previous sections; take $pi^e = \mathbb{1}[a^*(\theta^*)]$ and we recover the *benign* expert.
[8]The derivation follows the same step as in the proof of Proposition 1.

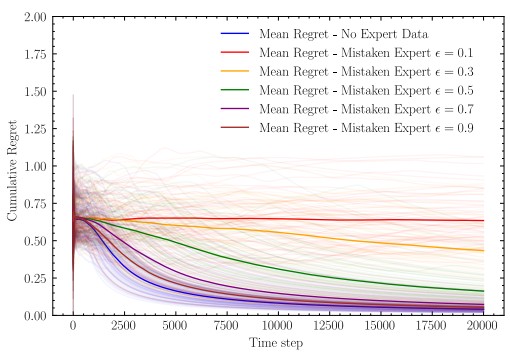 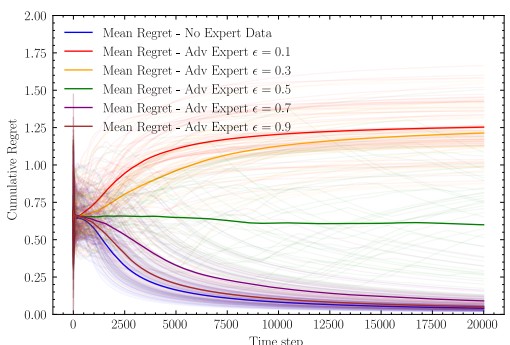

(a) Regret when learning from mistaken expert data.     (b) Regret when learning from adversarial expert data.

Figure 4: Regret obtained by TS agents with mistaken or adversarial expert data.

**Results for Adversary Experiments**    As discussed in Section C.1, we can see how the mistaken expert, in the worst case, induces no improvement of regret, which is reasonable since it samples from all actions uniformly. As $\epsilon$ increases, the only minimiser in $\Theta_q$ becomes $\theta^*$ since this is an asymmetric bandit class. Then, the cumulated regret still converges to zero, but at a much slower rate. For the adversarial expert results, we can see how for low $\epsilon$ the regret actually increases away from the mean 'uninformed' initial value; the expert forces the agent to believe it lives in a completely different world $\theta^{adv}$. Similarly, as $\epsilon$ increases, the set $\Theta_q$ becomes a singleton ($\theta^*$) and the agent still manages to achieve zero regret.

## C.3   Learning to Model the Expert

The existence of imperfect experts motivates the following question to be answered in this work: How should an online Bayesian learner estimate the expertise (or trustworthiness) of supposed experts, and how to incorporate this into their learning algorithm. Methods based on opponent modeling provide approaches for online Bayesian learners to estimate the behavior of other agents based on observations. Therefore, we conjecture that a solution to the possibility of experts being imperfect is to *infer the expert's policy based on their outcomes*. We describe first how such approach would be incorporated into the TS agents, and analyse afterwards the implications and caveats of such inference procedures.

**Beliefs over Expert Policies**    We consider the following assumptions. The expert has a (static) policy $\pi_e \in \Delta(\mathcal{A})$ from which they sample actions. Importantly, we assume that the expert's capabilities to corrupt samples $Y_t^e$ are limited; they cannot fabricate outcomes and are constrained to sampling from some (true) outcome distribution $p_a^*$. We assume therefore that the expert samples $A_t^e \sim \pi_e$, and the learner observes $Y_t^e \sim p_{A_t^e}^*$. A solution to handle the uncertainty over the expert policy $\pi_e$ is to keep a (history dependent) joint prior $P_t(\theta, \pi_e)$, assuming it factorises as $P_t(\theta, \pi_e) = P_t(\theta)P_t(\pi_e)$, and $P_t(\pi_e)$ being in the form of a Dirichlet distribution with $P_0(\pi_e) = \text{Dir}(\eta_0)$ and $\eta_0 \in \mathbb{R}_+^{|\mathcal{A}|}$ are the prior action-indexed Dirichlet parameters. Under policy $\pi_e$, the marginalised likelihood of a sampled expert outcome $Y_t^e$ is $P_t(Y_t^e \mid \theta, \pi_e) = \sum_a \pi_e(a)p_{\theta,a}(Y_t^e)$. Then, for realisation $Y_t^e$, the joint posterior update would be

$$P_{t+1}(\theta, \pi_e) \propto P_t(Y_t^e \mid \theta, \pi_e)P_t(\theta)P_t(\pi_e).$$

Note that this update is in general intractable; it couples the beliefs over $\theta$ and $\pi_e$, and separating them and updating them sequentially could lead to biases and incorrect updates. However, one can still approximate this posterior via a collection of particles $\{(\theta^{(k)}, \pi_e^{(k)}, w_t^{(k)})\}_{1 \leq k \leq K}$, and updating the weights for a collected sample $Y_t^e$ via their likelihood and renormalising[9].

Observe that holding beliefs over expert policies induces a new feature when learning from expert samples; in this case, the expert samples convey information about *both* the parameter $\theta$ and the expert

---

[9]We assume for the sake of the analysis to come that this approximation is tractable and accurate for the problems considered.

policy $\pi_e$, while a self-collected sample $(A_t, Y_t)$ conveys information only about $\theta$. Regardless of this, the most important unknown to gain certainty from information remains the same: $P_t(A^* = a)$. And in particular, we can use the same quantity to decide what source to learn from: the MI between $(A^*, Y_t^e)$.

**Estimating the MI with Expert Uncertainty**  We will now rely on the assumption that the prior factorises over $\pi_e$ and $\theta$ and on the fact that (either from a Dirichlet distribution or the particle weights) we can compute $\mathbb{E}_{P_t(\pi_e)}[\pi_e(a)] = \bar{\pi}_e(a)$. First, recall the MI can be written in terms of KL divergences:

$$\mathbb{I}_t(A^*, Y_t^e) = \sum_{a' \in \mathcal{A}} \pi_t(a') D_{KL}(P_t(Y_t^e \mid A^* = a') \| P_t(Y_t^e)).$$

We simply need then estimate $P_t(Y_t^e \mid A^* = a')$, $P_t(Y_t^e)$ to compute the corresponding MI and compare it with the MI computed from self-collected data in (6). First,

$$P_t(Y_t^e) = \mathbb{E}_{P_t(\theta, \pi_e)} \Big[ \sum_a \pi_e(a) p_{\theta, a}(Y_t^e) \Big] =$$

$$= \sum_a \bar{\pi}_e(a) \mathbb{E}_{P_t(\theta)}[p_{\theta, a}(Y_t^e)] = \sum_a \bar{\pi}_e(a) P_t(Y_t^e \mid A_t^e = a),$$

where the first equality holds from the independence assumption and $P_t(Y_t^e \mid A_t^e = a) \equiv P_t(Y_t \mid A_t = a)$; it is simply the probability of observing outcome $Y_t^e$ having selected action $A_t^e = a$. Now similarly, for $P_t(Y_t^e \mid A^* = a')$:

$$P_t(Y_t^e \mid A^* = a') = \frac{\mathbb{E}_{P_t(\theta, \pi_e)}\Big[ \mathbb{1}[\theta \in \Theta_{a'}^*] \sum_a \pi_e(a) p_{\theta, a}(Y_t^e) \Big]}{P_t(A^* = a')} =$$

$$= \sum_a \bar{\pi}_e(a) \frac{\mathbb{E}_{P_t(\theta, \pi_e)}\Big[ \mathbb{1}[\theta \in \Theta_{a'}^*] p_{\theta, a}(Y_t^e) \Big]}{P_t(A^* = a')} =$$

$$= \sum_a \bar{\pi}_e(a) P_t(Y_t^e \mid A_t^e = a, A^* = a').$$

With these, the learner can estimate the MI from observing expert data $\mathbb{I}_t(A^*, Y_t^e)$, which *incorporates the uncertainty about $\pi_e$*, and compare it against $\mathbb{I}_t(A^*, (A_t, Y_t))$ to decide which information source to process. For an algorithm to solve this expert modeling problem, see Algorithm 2.

---

**Algorithm 2** Information Choice with Trust Inference

---

1: Initialize particles $\{(\theta^{(k)}, \pi_e^{(k)}, w_0^{(k)}\}_{k=1}^K$ from priors $P_0(\theta), P_0(\pi_e)$.
2: **for** $t = 1, ..., T$ **do**
3:      Compute agent's policy $\pi_t(a) = P_t(A^* = a)$ using particles.
4:      Estimate self-play MI, $I_s \leftarrow \mathbb{I}_t(A^*; (A_t, Y_t))$, using $\{\theta^{(k)}, w_t^{(k)}\}$.
5:      Estimate expert MI, $I_e \leftarrow \mathbb{I}_t(A^*; Y_t^e)$, using joint particles $\{(\theta^{(k)}, \pi_e^{(k)}, w_0^{(k)}\}_{k=1}^K$. ▷ Can be done through sampling outcomes.
6:      **Decision:** $d_t \in \arg\max_{d \in \{s, e\}} \{I_s, I_e\}$
7:      **if** $d_t = s$ **then**                                            ▷ Learn from self-play
8:          Sample action $A_t \sim \pi_t$ and observe outcome $Y_t$.
9:          Update $P_{t+1}(\theta)$ using likelihood $p_{\theta, A_t}(Y_t)$.
10:      **else**                                                          ▷ Learn from expert
11:          Observe expert outcome $Y_t^e = y$.
12:          Compute $L^{(k)} = \sum_{a \in \mathcal{A}} \pi_e^{(k)}(a) p_{\theta^{(k)}, a}(y)$.
13:          Update posterior weights $w_{t+1}^{(k)} \propto w_t^{(k)} L^{(k)}$ and renormalise.
14:      **end if**
15:      Resample particles if necessary.
16: **end for**

---

## C.4 Modeling the Expert: When to Trust

We simulate symmetric and strongly asymmetric scenarios with $M = 500$ models and $|\mathcal{A}| = 20$ actions to reduce the particle sampling requirements. We simulate three agents; one agent learns from its own data, a second agent assumes *the expert is optimal and truthful* and learns from the maximum MI source, and a third agent *models the expert* and uses Algorithm 2 to decide what source to learn from. Different to previous experiments, here the expert is *boundedly rational*: it samples the optimal action with probability $\epsilon = 0.5$, or a randomly sampled action with probability $1 - \epsilon = 0.5$. *Results for MI estimating Agents* The results in Figure 5 allow us to distill two main conclusions.

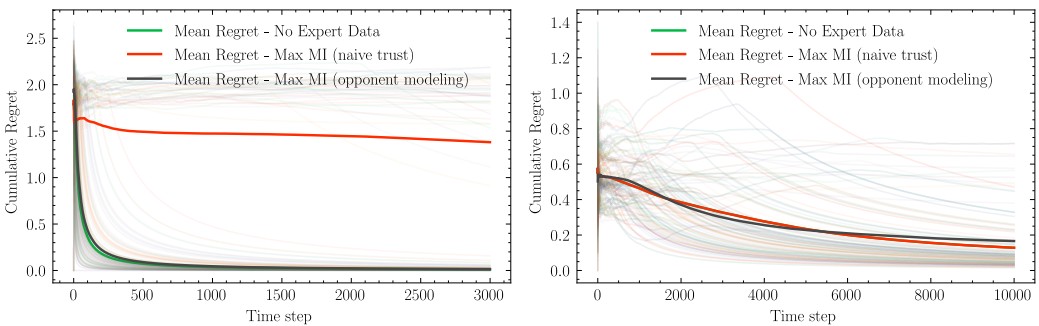

Figure 5: Regret obtained by MI estimating agents in symmetric (right) and asymmetric bandits (left).

First, in symmetric bandits where expert information provides no gain, all agents perform similarly; both MI estimating agents are able to discard expert information since it provides no additional knowledge over the environment, regardless of the expert modeling misspecifications. Second, for the asymmetric problem, the naive trust agent incurs high sustained regret; it is attempting to update its posterior assuming the expert is optimal, but the expert is in fact sampling different actions, leading to model misspecifications. Interestingly, the opponent modeling agent learns to identify this, and selects their own samples as information source, confirming our hypotheses and motivations.

