# OpenReview forum: "Bayesian Decision Making around Experts"
_NeurIPS.cc/2025/Workshop/Reliable_ML — NeurIPS 2025 - Reliable ML Workshop_

### Official Review · Reviewer_QCUh · 2025-09-19
**This paper studies Bayesian learning in multi-armed bandits with access to an expert's outcomes (not actions). They explore when to listen to to the expert and propose a particle-based algorithm to choose between expert and self information by maximizing one-step mutual information. The algorithm is not robust to bias and variance issues as discussed in the paper, and the robustness analysis is limited.**

**Rating:** 7
**Confidence:** 3

**Review:**

This paper studies Bayesian multi-armed bandits in the setting where, in addition to the learner’s own action–outcome data, outcomes from an expert’s optimal arm are available. The authors formalize how such expert outcomes can be incorporated into Bayesian posteriors, show that offline expert data reduces regret bounds by an amount proportional to the mutual information exchanged between expert data and the optimal action distribution, and present a particle-based algorithm to choose between expert and self information by maximizing one-step mutual information in the simultaneous setting. They also analyze the risks of mistaken or adversarial experts and provide experiments showing that expert data is most beneficial in asymmetric environments.

Strengths
- Prior work on expert advice in bandits focuses on actions, whereas this paper studies the effect of outcomes from an expert’s optimal arm.
- They link regret reduction and mutual information.
- Empirical results support the theory and illustrate when expert data is (or is not) useful.
- Interesting read.

Weaknesses
- The main assumption that the expert always plays the optimal arm is strong, and the robustness analysis for imperfect experts is limited.
- As discussed in the paper, the presented algorithm is fragile with high variance and potential learning collapse in symmetric settings.

Suggestions for Authors: Experiments on more complex bandit models could strengthen the practical impact.

---

### Official Review · Reviewer_5ukL · 2025-09-23
**Review of Bayesian Decision Making Observing an Expert**

**Rating:** 7
**Confidence:** 3

**Review:**

**Summary**

The paper expands the usual Thompson sampling framework to include information from experts. The paper considers two models of learning from experts. In the first, the expert provides data from the optimal reward distribution to the learner. The learner uses this data to update their prior. In this setting, they provide an upper bound on the learner's regret that decreases with increase in mutual information between the expert data and the optimal-arm's distribution. In the second model, the learner receives the datatset from the expert sequentially rather than all at once. And the learner is only allowed to use either their own samples which includes information about the arm it comes from or the expert's sample which does not include information about the arm, but is guaranteed to be from the optimal arm. The paper also considers some models of misspecification of the expert's information i.e., models of how the expert's data can differ from exact draws from the optimal distribution. In simulations, they check how these misspecifications can lead to a lack of robustness.

**Strengths**

The problem is a natural formulation of a very interesting problem of utilizing expert data. The decision problem of choosing between expert and learner's own data has some interesting tradeoffs that the paper motivates.

The dependence of the regret bound in the experts setting clearly elucidates an interpretable quantity of mutual information of expert's data as controling the benefit of the expert.

**Weaknesses**

While the sequentual update setting where the learner faces the decision problem of which data to use to update beliefs is interesting, there don't seem to be a lot of insights provided. There are no guarantees or nice properties of the proposed algorithm stated. There are also no insights into the decision problem i.e., when learner's data is preferred or when expert's data is preferred.

**Suggestion**
Stating more guarantees, perhaps from experimental design, to justify Algorithm 1 would be helpful.

In the experiments, it would be helpful to show when learner's data is preferred vs expert's data is preferred.

---

### Official Review · Reviewer_BMpF · 2025-09-25
**Review of Bayesian Decision Making Observing an Expert**

**Rating:** 8
**Confidence:** 4

**Review:**

This paper studies two settings of bandit problems where the player is equipped with the ability to observe the outcomes of an expert but not their actions. In the offline setting, the player observes a sequence of outcomes of an expert before interacting with the environment. In the simultaneous learning setting, the player at each time step chooses an action, and then may choose to observe either the outcome of their own chosen action, or the outcome of the expert. The main result gives an improved bound to the regret in the offline setting compared to the vanilla Thompson Sampling algorithm, where the gain is quantified by the mutual information between the observed outcomes of the expert and the optimal arm. The paper also outlines a few problems for future work in the simultaneous setting and in generalizing the result to continuous parameter space. I think the problem introduced is novel, and audience from broad learning theory would find the result interesting.

Pros: The problem is well formulated and accompanied the theoretical framework and results with empirical justification. Further work is also outlined in the paper with clear motivation to tackle them.

Cons: I wonder if the upper bound in Theorem 1 is tight.